# Electron-Induced Decomposition of 5-Bromo-4-thiouracil and 5-Bromo-4-thio-2′-deoxyuridine: The Effect of the Deoxyribose Moiety on Dissociative Electron Attachment

**DOI:** 10.3390/ijms24108706

**Published:** 2023-05-13

**Authors:** Farhad Izadi, Adrian Szczyrba, Magdalena Datta, Olga Ciupak, Sebastian Demkowicz, Janusz Rak, Stephan Denifl

**Affiliations:** 1Institut für Ionenphysik und Angewandte Physik, Universität Innsbruck, Technikerstrasse 25, A-6020 Innsbruck, Austria; 2Center for Molecular Biosciences Innsbruck, Universität Innsbruck, Technikerstrasse 25, A-6020 Innsbruck, Austria; 3Laboratory of Biological Sensitizers, Department of Physical Chemistry, Faculty of Chemistry, University of Gdańsk, Wita Stwosza 63, 80-308 Gdańsk, Poland; 4Department of Organic Chemistry, Faculty of Chemistry, Gdańsk University of Technology, Narutowicza 11/12, 80-233 Gdańsk, Poland

**Keywords:** BrSU, BrSdU, uracil derivatives, low-energy electron attachment, DEA

## Abstract

When modified uridine derivatives are incorporated into DNA, radical species may form that cause DNA damage. This category of molecules has been proposed as radiosensitizers and is currently being researched. Here, we study electron attachment to 5-bromo-4-thiouracil (BrSU), a uracil derivative, and 5-bromo-4-thio-2′-deoxyuridine (BrSdU), with an attached deoxyribose moiety via the N-glycosidic (N1-C) bond. Quadrupole mass spectrometry was used to detect the anionic products of dissociative electron attachment (DEA), and the experimental results were supported by quantum chemical calculations performed at the M062X/aug-cc-pVTZ level of theory. Experimentally, we found that BrSU predominantly captures low-energy electrons with kinetic energies near 0 eV, though the abundance of bromine anions was rather low compared to a similar experiment with bromouracil. We suggest that, for this reaction channel, proton-transfer reactions in the transient negative ions limit the release of bromine anions.

## 1. Introduction

Radiotherapy is among the most commonly used treatments for cancer though the tumor environment’s specificity, such as a low concentration of oxygen (hypoxia), limits the efficacy of this modality [1,2,3]. Hypoxia, a feature of solid tumors, induces tumor cells to develop resistance to ionizing radiation (IR) [4]. IR is used in radiation therapy to deposit energy in the biological medium. As the radiation passes through the medium, a large portion of the deposited energy is channeled into the generation of low-energy electrons (LEEs) with kinetic energies less than a few hundred eV and an energy distribution maximum of approximately 9–10 eV [5]. When DNA is exposed to so-called ballistic electrons, the latter are known to induce DNA damage such as single- and double-strand breaks, base release, and sugar modifications upon dissociative electron attachment (DEA) [6,7,8,9]. However, electrons are just not detrimental to hydrated DNA with respect to strand breaks [10,11]. Given that solid tumor cells are hypoxic, they are also resistant to hydroxyl radicals (^●^OH), a significant harmful agent of native DNA generated during radiotherapy [12], additionally to LEEs. To be effective, radiotherapy should be combined with radiosensitizers, which are compounds that can sensitize cells to ionizing radiation. However, according to the current clinical trials, the number of radiosensitizers being evaluated is relatively low [13]. The circumstance is even worse when it comes to clinically permitted radiosensitizers. For example, no radiosensitizers are utilized in medical care against gastrointestinal cancers [14].

Nimorazole, 4-[2-(5-nitroimidazol-1-yl)ethyl]morpholine, is a brilliant example of a radiosensitizing molecule that has been approved to treat head and neck cancers [15] but is only used in Nordic countries. Uridine analogs that are incorporated into DNA are radiosensitizers that include the most thoroughly studied 5-bromo- and 5-iodo-2′-deoxyuridines [16]. They are phosphorylated in the cytoplasm to produce 5-triphosphates before being incorporated into cellular DNA by human DNA polymerases [17]. Being a component of DNA, they are prone to the electron-attachment-induced dissociation of the C5-X bond, which produces a halide anion and leaves behind a reactive uracil-5-yl radical in the biopolymer molecule [18]. Although numerous studies suggest that the increased level of damage and cell death is caused by dissociative electron attachment to DNA labeled with BrdU or IdU, none of the compounds is presently used in clinics [15,19]. Thus, other radiosensitizers should be studied for a possible clinical application.

Regarding the basic molecular mechanisms of radiosensitizers, DEA may be exploited, in which an electron is resonantly captured at a particular electron energy, leading to the formation of a transient negative ion (TNI). The TNI is usually unstable in terms of electron detachment or dissociation. In the latter process, the TNI dissociates into an anion and neutral radical [20,21].
e^−^ + AB → (AB)*^−^ → A^−^ + B(1)

Here, AB is the parent molecule, (AB)^●−^ represents the TNI, while A^−^ and B stand for the fragment anion and the corresponding neutral species, respectively. It is worth noting that DEA can be extremely effective at electron energies near zero eV (see results section of this study) in comparison to the spontaneous emission of the excess electron. Consequently, in more recent studies, it has been demonstrated that the presence of high-electron-affinity substituents in nucleobases makes them more susceptible to DEA. Rak and coworkers proposed a number of 5-substituted pyrimidine derivatives as potential radiosensitizers of hypoxic cells. Such compounds, which include 5-trifluoromethanesulfonyl-uracil (OTfU), uracil-5-yl O-sulfamate (SU), and 5-selenocyanatouracil (SeCNU), have been researched theoretically and experimentally through stationary radiolysis in an aqueous solution and single-molecule studies in the gas phase [22,23,24]. Earlier, the favorable attachment behavior had already been observed in thoroughly researched halogenated uracil derivatives [25,26,27,28] such as 5-bromouracil (BrU). Further studies showed that 5-bromouridine (BrdU), similar to halouracils, is very sensitive to low-energy DEA, with Br- as the dominant product [29]. Recently, it has been shown that ISdU (5-iodo-4-thio-2′-deoxyuridine), a substituted uridine, increases the extent of tumor cell killing with ionizing radiation. Amazingly, a similar derivative of 4-thio-2′-deoxyuridine, 5-bromo-4-thio-2′-deoxyuridine (BrSdU), lacks radiosensitizing properties in cell culture studies [30]. Further radiolysis studies of BrSdU combined with quantum chemical calculations indicated that BrSdU lacks favorable DEA characteristics in the solution phase. The B3LYP/6-31++G(d,p) kinetic barriers for breaking the C5-Br and C5-I bond are equal to 0.27 and 0.13 eV, respectively. The abovementioned barriers suggest thus that, at ambient temperature, the lifetime of the BrSdU radical anion is nearly 200-fold longer than that of ISdU. Such a significant lifetime difference probably enables BrSdU to be protonated before the actual C5-Br bond dissociation, and protonation, in turn, may prevent dissociation due to a significant increase in the barrier for C5-Br cleavage [31]. Thus, a higher activation barrier for DEA leads to the quenching of DEA and probably to protonation of the TNI [30].

In order to investigate whether the quenching of DEA to BrSdU is a particular effect of the solution phase and not an intrinsic property of the molecule, we performed electron attachment studies with the isolated compound in the gas phase. Herein, we report the results of crossed electron–molecular beam experiments on the anion formation and fragmentation pathways prompted by the attachment of electrons with kinetic energies between ~0 and 10 eV. Since, during the experiments, it turned out that BrSdU is prone to thermal decomposition upon transfer into the gas phase by thermal heating, we experimentally focused on 5-bromo-4-thiouracil (BrSU), a derivative of uracil. Quantum chemical calculations on the thermodynamic thresholds of fragmentation reactions for BrSU support the experimental findings. In spite of the experimental difficulties with BrSdU, we also calculated the thermodynamic thresholds of DEA reactions for this molecule. A comparison of the computational results for the nucleobase and the nucleoside species allows the investigation of the influence of the deoxyribose ring on the DEA process.

## 2. Results and Discussion

### 2.1. Formation of BrSU^−^ Anions and the Dehydrogenated (BrSU-H)^−^ Anions—Cleavage of the Br-C5 Bond

The anion efficiency curve for the parent anion of BrSU (*m*/*z* 206) is shown in Figure 1a. It is formed most abundantly in a peak at ~0 eV. Similar behavior was observed for BrU, which led to the formation of the corresponding parent anion BrU^−^ [32] as well as for BrdU, which also showed the formation of a detectable parent anion [29]. From the anion efficiency curve shown in Figure 1a, we derived another low-intensity peak at 0.55 eV. These peak positions are also listed in Table 1, which summarizes all the peak maxima of the detected anions from DEA to BrSU. The raw data of the measurements are included in the Appendix A (BrSU-BrSdU-Raw data.xlsx file). The proposed reaction pathways for anions resulting from BrSU are shown in Figure 2. According to our calculations, the electron affinity (EA) of BrSU is positive and amounts to 1.18 eV. The positive EA agrees well with the experimental detection of the parent anion.

The anion yield of the dehydrogenated parent anion of (BrSU-H)^−^ (*m*/*z* 205) is depicted in Figure 1b and represents the following reaction (Figure 2, reaction 2a):e^−^ + BrSU → (BrSU)*^−^ → (BrSU-H)^−^ +H^⦁^
(2)

(BrSU-H)^−^ is detected in the first peak close to ~0 eV. A previous DEA study with uracil demonstrated that the N1-H site is the thermodynamically favored one for dehydrogenation [33]. According to our thermodynamic calculations, H abstraction from the N3-H site of BrSU is already an exothermic reaction with a threshold of −0.82 eV. From the thermodynamic aspect, we also investigated H-loss from the C6 site, which is, however, endothermic by 1.54 eV (Figure 2, reaction 2b).

DEA to 2-thiouracil (TU) and 1-methyl-2-thiouracil was studied by Kopyra and co-workers [32]. The 0 eV peak observed for (TU-H)^−^ production was proposed in ref. [32] to be initiated by the capture of the excess electron into the $\pi_{1}^{*}$ orbital (shape resonance) [34]. In addition to shape/core-excited resonances, it has been demonstrated that dipole-bound anions (DBAs), i.e., the attachment of the electron by the molecule’s long-range dipolar field [35,36,37], can be a major gateway for DEA via vibrational Feshbach resonances (VFRs) [38,39,40].

In this case, DEA occurs if the dipole-bound state couples to some dissociative valence state. For the canonical nucleobases uracil and thymine, the antibonding $\sigma_{N-H}^{*}$ was suggested to result in the formation of the dehydrogenated parent anion [41,42]. On the other hand, in thymine and uracil, the H atom may be also removed through the cleavage of the N3-H, C5-H, and/or C6-H bonds. However, previous studies with uracil and thymine [43] found no formation of the dehydrogenated parent anion by C-H bond cleavage despite the fact that it was thermodynamically accessible at higher energy [40,44]. As a result, the dehydrogenation of the nucleobase anion must result merely from the rupture of the N-H bond.

From the anion efficiency curve for (BrSU-H)^−^ shown in Figure 1b, we deduced five features: in addition to the peak at ~0 eV, two higher-intensity peaks at 0.2 eV and 0.5 eV are detected, which grow in a sequence and are followed by weaker shoulders near 0.8 and 1.3 eV. All these features lie below the calculated threshold for H-loss from the C-6 position (Figure 2, reaction 2b). Thus, it can be excluded that the H-loss from the carbon site contributes to the ion yield.

In the present study, the fragment anions formed by cleavage of the Br-C5 bond were the most abundant ones. However, for the present case, it was the radical Br^⦁^ that was formed predominantly along with the respective ion (BrSU-Br)^−^ at *m*/*z* 127 (Figure 2, reactions 3a and 3b),
e^−^ + BrSU → (BrSU)*^−^ → (BrSU-Br)^−^ + Br^⦁^(3)

(BrSU-Br)^−^, in this context, represents the fragment anion formed by the release of the bromine radical from the TNI. The corresponding molecular structure of this fragment anion is (C_4_H_3_N_2_SO)^−^. The anion yield is shown in Figure 1c and reveals three features for this anion, dominated by a sharp peak at ~0 eV, another resonance at 0.17 eV, and a broad bump around 0.54 eV. We note that, in the previous DEA study with BrU, three peaks at ~0 eV (major peak), 1.4 eV, and 6 eV were observed in the (BrU-Br)^−^ anion yield [45]. Therefore, the yields for bromine release from BrSU and BrU just share the abundant peak near zero eV. Our calculated thermodynamic threshold for bromine release from BrSU was −1.03 and −0.44 eV, as calculated at the M062X/aug-cc-pVTZ level. These processes are related to proton transfer from N1-H to the C5 site or from N3-H to the C5 site, respectively, in the BrSU radical anion followed by the bromine atom release (Figure 2, reaction 3a), as suggested for BrU in ref. [46]. The thermodynamic threshold for the Br atom release proceeding without the abovementioned proton transfer (Figure 2, reaction 3b) amounts to as much as 1.17 eV. No peak is observed in the (BrSU-Br)^−^ ion yield above this energy, and thus, this channel can be excluded. Here, one may ask how 0 eV electrons lead to a process associated with high-barrier proton transfer. It is worth noticing that the EA of BrSU amounts to 1.18 eV (Table 1), and the experiment is carried out in a single-collision regime, which means that the excess energy originating from electron attachment is not dissipated in the collisions with molecules from the surrounding. In order to propose a possible proton-transfer mechanism, we calculated an energy profile for the stepwise proton transfer between neighboring proton-acceptor sites in the BrSU radical anion. A simpler, direct proton transfer between N1 and C5 is not probable since, in the respective transition state, the proton would be almost completely detached from N1 while a new bond (C5-H) would not be formed. Therefore, we assumed the following proton-transfer sequence: N1 to O2, N3 to S4, O2 to N3, and S4 to C5. Already, the first step, the N1 to O2 transition, is associated with an energy barrier of 1.80 eV. Thus, 1.18 eV, originating from the electron attachment to BrSU, is not sufficient for the N1 to O2 proton transfer. One should note, however, that the theoretical threshold due to proton transfer from N1 to C5 (−1.03 eV; Table 1 and Figure 2—reaction 3a) reproducing the experimental result may not be the only one. Indeed, a theoretical threshold associated with the N3 to C5 transition is equal to −0.44 eV (see Table 1 and Figure 2—reaction 3c), which still agrees with the 0 eV threshold measured experimentally. What is more important, the respective tautomer (reaction 3c in Figure 2) may form in a two-step proton-transfer process: N3 to S4 and S4 to C5, coupled with the activation barriers of 1.41 and 1.02 eV, respectively. Finally, the energy difference between the second transition state and the BrSU^−^ radical anion amounts to 1.63 eV. Although the EA of 1.18 eV is somewhat smaller (0.45 eV) than the latter value, this discrepancy may result from the employed theoretical model (an average error for EA prediction at the M06-2X level amounts to as much as 0.19 eV [47], while that for thermochemistry and kinetics amounts to ca. 0.06 eV [46]) and 0.1 eV resolution of electron energy in the experiment. All these uncertainties account for an error of at least 0.35 eV that could lower the abovementioned difference to ca. 0.1 eV.

Another abundant fragment anion accompanied by cleavage of the Br-C5 bond was detected at *m*/*z* 126 and is assigned to (BrSU-HBr)^−^ (Figure 2, reactions 4a, 4b, and 4c)
e^−^ + BrSU → (BrSU)*^−^ → (BrSU-HBr)^−^ + HBr (4)

This fragment anion appears as the second most abundant one in the experiment (see Figure 1d). The anion yield curve exhibited two narrow peaks at ~0 and 0.25 eV, followed by a broad bump near 0.7 eV. We computationally investigated reaction (4) by considering H abstraction from the N1 position (Figure 2, reaction 4a). The calculated thermodynamic threshold predicts an endothermic reaction with a threshold of 0.21 eV. Taking into account the accuracy of the calculations (mean unsigned error for thermochemistry amounts to about 0.06 eV [48] for the assumed level of theory) and the vibrational excitation of the neutral molecule at the used sublimation temperature, the agreement is reasonable. The loss of hydrogen from the N3 nitrogen or C6 carbon (Figure 2, reaction 4c and 4b) is associated with a threshold of 0.70 and 1.27 eV, respectively. Below the first electronically excited state for uracil, estimated at ~5.0 eV [49], the transitory anion can be formed through one of the following two mechanisms: (1) shape resonance (occupation of a $\pi_{}^{*}$ molecular orbital by the extra electron and subsequent coupling to dissociative $\sigma^{*}$ state) [50] or (2) vibrational Feshbach resonances (VFRs) [51]. Differences in the dissociation mechanism may then serve as an explanation for the relative abundances of peaks observed; for example, the broad bump above 0.5 eV is more pronounced for (BrSU-HBr)^−^ than for (BrSU-Br)^−^.

Figure 3 presents the anion efficiency curve for the formation of the anion at *m*/*z* 79 upon DEA to BrSU. It is obvious to assign this yield to the single-bond cleavage reaction (Figure 2, reaction 5),
e^−^ + BrSU → (BrSU)*^−^ → Br^−^ + (BrSU-Br)^⦁^(5)

We obtained two sharp peaks at ~0 eV and 0.4 eV, followed by far weaker signals between 1 and up to about 7 eV. The calculated thermodynamic threshold signifies an endothermic reaction with a threshold of 0.13 eV. This threshold would be consistent with the onset of the main peak observed at 0.4 eV, while the sharp peak near zero eV may correspond to DEA to vibrationally excited neutral BrSU. The same fragment anion was observed upon DEA to BrU [27,45] in a strong zero-eV resonance. Different energy resolutions of the electron beam may account for the difference that just one single zero-eV peak was obtained in [27,45]. However, it is interesting to note that, contrary to previous observations for BrU [27,45], we discovered Br^−^ as only the third most abundant anion for BrSU. A tentative explanation for the low abundance of Br^−^ may be the competition to reaction 3a (complementary reaction with the excess charge localized on the nucleobase moiety). Reaction 3a is exothermic, i.e., accessible for electrons with energies of ~0 eV. The reciprocal dependence of the electron attachment cross section on the electron energy may lead to the high yield [52], though the proton-transfer reaction in the (BrSU-Br)^−^ anion must occur as well. Another competitive channel of Br^−^ abstraction would be the release of HBr, reaction 4. In this case, electron attachment to BrSU may lead first to a TNI state with the excess charge mainly localized at the bromine atom. Before the release of the atomic anion happens, a proton transfer from the neutral moiety leads to the formation of the HBr molecule and (BrSU-HBr)^−^. Such a mechanism was previously proposed for fluorouracil, in which the formation of F^−^ was weak compared to the release of neutral HF molecules [53,54].

### 2.2. Light Anions Formed upon DEA to BrSU

In the course of the present study, we also observed two lighter fragment anions at *m*/*z* 42 and *m*/*z* 33 that can be formed through dissociation of BrSU upon DEA. Figure 4a,b depict the corresponding anion efficiency curves. The fragment anion at *m*/*z* 42 can be assigned to NCO^−^ (Figure 2, reaction 6a and 6b),
e^−^ + BrSU → (BrSU)*^−^ → NCO^−^ + (BrSU-NCO)(6)

Reaction (6) represents a more complex reaction involving multiple bond cleavages followed by rearrangements. The ion yield shown in Figure 4a shows two low-intense peaks at ~0 eV and 1.0 eV, followed by the main feature centering around 3 eV. Another abundant peak can be found near 5 eV. The earlier studies with BrU reported NCO^−^ formation as well [27,45]; however, the ion yield included a rather narrow, intense peak near 1.6 eV, which is absent in Figure 4a. For NCO^−^, two different sites of formation in the pyrimidine ring are possible: N1-C2=O and/or O=C2-N3 (see Figure 2, reactions 6b and 6a). The threshold of 1.13 eV was calculated for the former site, while it was 1.30 eV for the latter one. Thus, the main signals in Figure 4a are formed above this threshold, and the minor features near zero and 1 eV may be ascribed to other pathways.

The small fragment anion at *m*/*z* 33 may form due to the cleavage of the double bond S=C and the transfer of one hydrogen atom from another position in order to connect to SH^−^ (Figure 2, reaction 7),
e− + BrSU → (BrSU)*^−^ → SH^−^ + (BrSU-SH)(7)

Similar to the release of neutral HBr, H-atoms from different sites in the molecule may contribute. In the calculation, we considered H abstraction from the N3 position and obtained a threshold of 1.74 eV, reaction (7). Compared with the measured anion efficiency curve shown in Figure 4b, the computed threshold would match with the onset of the main peak at 2.4 eV. Two features quite similar to the NCO^−^ ion yield were observed at lower energies. Thus, those peaks may correspond to other dissociation reactions.

Finally, we note that Kopyra and Abdoul-Carime reported SCN^−^ and S^−^ formation in DEA to 2-thiouracil [55]. Remarkably, we did not observe signals for these two anions in our study with BrSU. Such a different result may be only explainable by the different position of the sulfur atom at the pyrimidine ring (C4 in the present molecule and C2 in the thiouracil). The chemistry induced by the initial electron attachment may then be initiated very locally and strongly depending on the electronic structure of the corresponding temporary negative ion.

### 2.3. Possible Dissociation Channels of BrSdU

Due to the experimental problems with thermal decompositions mentioned in Section 3, the measured energy scans for the observed fragment anions of BrSdU are not discussed here in detail. The ion yields cannot be unambiguously assigned to DEA to an intact BrSdU sample, and therefore, we show them in the Appendix A for the sake of completeness. The corresponding peak positions and the experimental thresholds are listed in Appendix A. Just to note, the three major anions are the same as for BrSU and show quite similar anion yields as well as ion yield ratios. If the sample thermally decomposed to BrSU, one may then also expect BrSU^−^, as presented in the results above, but its absence may be explained by internal excitation due to pick-up of the hydrogen atom after decomposition.

Our computational results for BrSdU are summarized in Figure 5 and Table 2. We focused on the anions formed in the experiment. Moreover, the calculations predict that the parent anion of BrSdU (EA = +1.28 eV), reaction A in Figure 5, is slightly more stable than that of BrSU. If we look at the most important dissociation reaction with reference to radiosensitization, we obtain a threshold of 0.14 eV for the release of Br^−^ and the neutral radical (BrSdU-Br)^⦁^ (see Figure 5, reaction 6). This threshold is very close to that for BrSU (0.13 eV). If we consider localization of the excess charge at the nucleobase moiety, the threshold is 1.00 eV (Figure 5, reaction B). In the case of additional cleavage of the glycosidic bond (this nominally corresponds then to the (BrSU-HBr)^−^ anion), we obtain a threshold of 0.31 eV and 1.23 eV; see Figure 5, reactions 4a and 4b, respectively. Both values also include bond formation between the bromine atom and the sugar moiety, which are both released from the negative ion. Then, the threshold is just slightly higher than that for HBr release from BrSU (0.21 eV).

The ambiguity of the experimental results is further indicated by the threshold calculations for the release of the (BrSU-Br)^−^ anion (*m*/*z* 127). To nominally form this anion from BrSdU, the C5-Br bond and the glycosidic bond must be broken with additional hydrogen migration to the base moiety (Figure 5, reactions 3a, 3b, and 3c).

Depending on the final positions of the three hydrogens at the nucleobase moiety, the threshold amounts to at least 0.45 eV (hydrogen atoms attached to N1, C6, and sulfur atom; Figure 5, reaction 3a), followed by 0.66 eV (N3, C6, and sulfur; Figure 5, reaction 3b) and 1.68 eV (N1, N3, and C6, Figure 5, reaction 3c). From these values, one may expect a different shape of the anion efficiency curve at *m*/*z* 127 for BrSdU and BrSU (threshold −1.03 eV, see above), which is not the case. On the other hand, thresholds for the light fragment anion NCO^−^ can be obtained in the case of BrSdU, which are not substantially different from those for BrSU. The deviation is just 20 meV, if the closure of the pyrimidine ring after excision of NCO^−^ is considered (see Figure 5, reactions 7a). For SH^−^ formation, the thresholds are almost identical as well (1.70 eV for BrSdU and 1.74 eV for BrSU). We also calculated the thresholds for two anions not found for BrSU (*m*/*z* 98 and *m*/*z* 190, see Figure 5, reactions 5 and 2). However, substantially endothermic threshold values were obtained (≥1.40 eV), while the experimental results showed zero-eV peaks (see Appendix A). The ion at *m*/*z* 98 is assigned to the localization of the excess charge at the sugar moiety.

## 3. Materials and Methods

### 3.1. Experiments

The experimental results presented in this paper were acquired using a crossed electron–molecular beam experiment that has been previously described in detail [6]. We only provide a brief description of this experiment here. A homemade hemispherical electron monochromator (HEM) was used to create a well-defined electron beam. Additionally, a copper oven with glass inset as a sample container, a quadrupole mass analyzer (QMA), and a detector to count the mass-analyzed ions comprised the setup. The beam of neutral molecules was formed by heating the sample in the oven and guiding the vapor through a 1mm diameter capillary into the interaction zone with low-energy electrons. The BrSU was synthesized according to the procedure described by Łapucha [56] (Appendix A), while the BrSdU sample was synthesized via the procedure described by Spisz et al. [30] (Appendix A). An electron beam with an energy resolution of ~100 meV at FWHM (full width at half maximum) was generated by the HEM. The used electron current was in the range of 25–38 nA. With such a current, the optimum balance between resulting ion beam intensity and electron energy resolution was achieved. A weak electrostatic field extracted the ions formed by (dissociative) electron attachment to the QMA entrance. The anions were mass-analyzed in the QMA and detected by a channeltron secondary electron multiplier operating in the single-pulse counting mode. A scan of the electron energy was used to track the anion efficiency curves, while the QMA was set for the transmission of a specific anion. The well-known Cl^−^/CCl_4_ resonance at 0 eV [57] was used to calibrate the electron energy scale and to determine the energy resolution. Using the method proposed in [58], the experimental thresholds for detected DEA reactions with both compounds were determined.

Before beginning the negative ion measurements, the temperature dependence of the electron ionization mass spectrum at the electron energy of ~70 eV was checked up to 383 K to ensure a suitable ion signal without sample decomposition in the oven. For nucleosides such as thymidine and uridine, the presence of the native nucleobase cation combined with a relative increase to other signals in the mass spectrum at higher temperatures was associated with thermal decomposition [59]. In this case, the glycosidic bond between sugar and base moieties breaks upon heating, and the base moiety picks up a hydrogen atom, forming the intact base. Thus, the results of negative ion formation for the decomposed nucleoside also resembled that of the nucleobase [60,61]. Regarding the cation mass spectrum of BrSdU, we indeed observed the analogous symptomatology, i.e., we observed a BrSU^+^ signal at *m*/*z* 206, which grew stronger with temperature than other signals and thus indicated the presence of thermally decomposed BrSdU. Subsequently, the energy scans of detected anions were measured up to the maximum temperature of 387 K.

For the studied BrSU sample, the same procedure was carried out as for BrSdU. Since the parent cation of BrSU showed up in the mass spectrum, it was easily traceable that no thermal decomposition product appeared in the studied temperature range up to 382 K. Subsequently, the energy scans of the detected anions were measured up to the maximum temperature of 389 K.

### 3.2. Quantum Chemical Calculations

In order to obtain the lowest-energy geometries of the neutral BrSU and BrSdU, as well as their anions and the respective molecular fragments, the unconstrained geometry optimizations were performed at the M06-2X [48]/aug-cc-pVTZ [55,62] level of theory (method/basis set) using restricted and unrestricted wavefunction for closed- and open-shell species, respectively. All such obtained geometries were geometrically stable, which was confirmed by the analysis of harmonic frequencies (all force constants were positive for minima, while all but one were negative for the first-order transition states). The activation barriers were calculated as the difference between the enthalpy of the transition state (TS) and the substrate. The intrinsic reaction coordinate (IRC) [63] procedure was used to verify that the transition state connects the proper minima. The energies of the optimized reactants were used for the calculations of thermodynamic thresholds related to the anions observed in the crossed electron–molecular beam experiments. The thresholds (ΔH) were calculated as the difference between the enthalpies of products and the neutral substrate in their electronic ground states. The enthalpies of the reactants result from correcting the relevant values of electronic energies for zero-point vibration terms, thermal contributions to energy, and the p,V term. These terms were obtained using the rigid rotor–harmonic oscillator approximation for T equal to 298.15, 385.15, 387.15, and 389.15 K. In the reaction studied within the experiment, electron energy is absorbed by a substrate that allows the system to move from the initial enthalpy level to that of the products. Therefore, the computational reaction enthalpy (thermodynamic threshold) should correspond to the electron energy, matching the onset of the experimental signal for the endothermic reactions. On the other hand, the computationally exothermic processes should be triggered by 0 eV electrons (an experimental thermodynamic threshold of 0 eV). All calculations were performed with the Gaussian 16 [64] suite of programs. 

## 4. Conclusions

In the present study, we investigated electron attachment to the modified nucleobase BrSU and the corresponding nucleoside BrSdU. For the latter molecule, we observed the indication of thermal decomposition during the thermal heating process, which leads to ambiguous experimental results. Alternative methods are needed to transfer this compound into the gas phase. For example, laser-induced acoustic desorption was reported to be a gentle method for the preparation of neutral targets in the gas phase [65]. Previous experiments with thermally labile molecules such as D-ribose-5-phosphate [61] have already demonstrated that this method may also be applicable for DEA experiments.

For BrSU, thermal decomposition did not play a role, and we observed strong decomposition of the molecule upon attachment of a low-energy electron with a kinetic energy of about zero eV. While cleavage of the Br-C5 bond turned out to be very efficient, as expected, the experimentally observed low abundance of the bromine anion was rather surprising. Instead, (BrSU-Br)^−^ was observed as the most abundant fragment anion. The present calculations of the thresholds for BrSU indicate that a proton transfer in the (BrSU-Br)^−^ fragment anion would lead to an exothermic threshold and explain the observed ion signals near zero eV. Such a proton-transfer reaction may also solve a contradiction for DEA to BrU in the literature, since experimentally a zero-eV peak was observed in the (BrU-Br)^−^ yield [27,45], while calculations of the threshold assuming the single Br-C5 bond cleavage predicted a high threshold of 1.25 eV [66]. In addition, the high abundance of (BrSU-HBr)^−^ may also indicate that subsequent proton transfer to the Br− in the transient negative ion may be another limiting factor for the release of the bromine anion. The calculations for BrSdU indicate almost the same endothermic threshold for the release of the bromine anion as that for BrSU, i.e., there is no significant influence of the deoxyribose moiety on the energetics of this channel.

If we consider BrSU as a model compound for radiosensitization studies in solution, this experimental result in the gas phase would support the limited effects observed in radiolysis studies with BrSdU [30]. In a water solution, proton-transfer reactions with surrounding molecules would even more effectively influence DEA reactions, as for example suggested in [67]. 

## Figures and Tables

**Figure 1 ijms-24-08706-f001:**
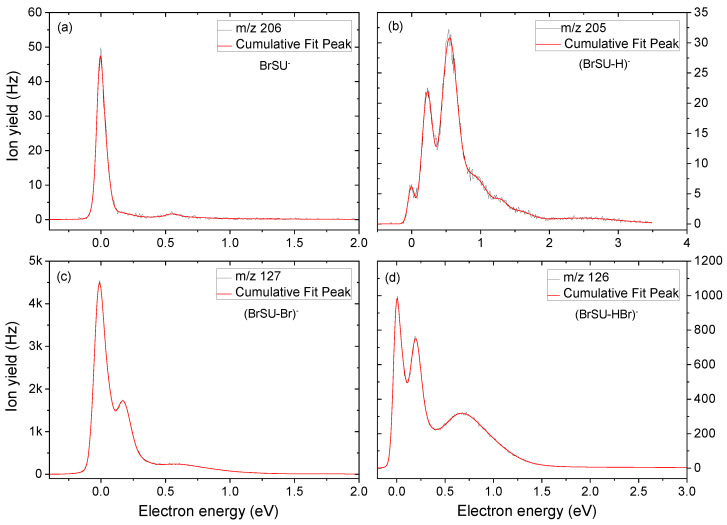
Anion efficiency curve for the fragment anions (**a**) BrSU^−^, (**b**) (BrSU-H)^−^, (**c**) (BrSU-Br)^−^, and (**d**) (BrSU-HBr)^−^ formed upon electron attachment to BrSU.

**Figure 2 ijms-24-08706-f002:**
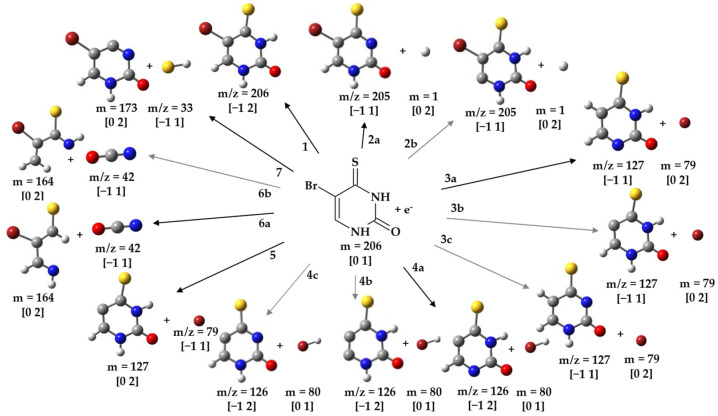
Possible DEA-induced reaction pathways for the degradation of BrSU; *m*/*z*, charge, and multiplicity (in the square brackets) are displayed near the particular structures. Thick arrows indicate the most abundant processes observed in the crossed electron–molecular beam experiment.

**Figure 3 ijms-24-08706-f003:**
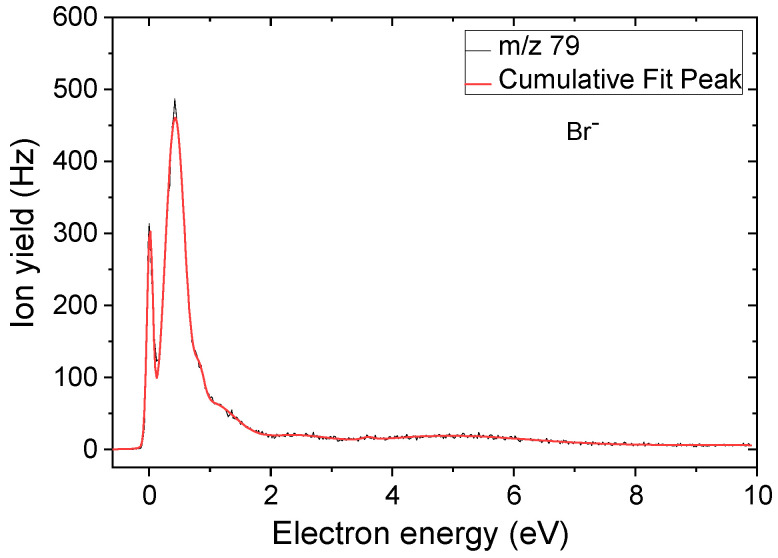
Anion efficiency curve for the Br^−^ fragment anion formed upon electron attachment to BrSU.

**Figure 4 ijms-24-08706-f004:**
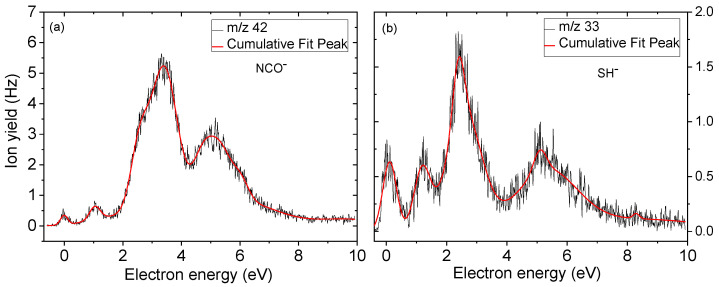
Anion efficiency curve for the fragment anions (**a**) NCO^−^, and (**b**) SH^−^ formed upon electron attachment to BrSU.

**Figure 5 ijms-24-08706-f005:**
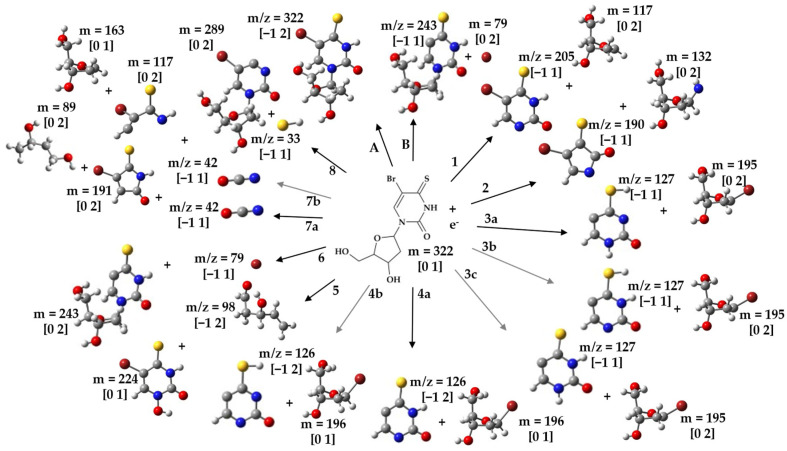
Possible DEA-induced reaction pathways for the degradation of BrSdU; *m*/*z*, charge, and multiplicity (in the square brackets) are displayed near the particular structures. Thick arrows indicate the most abundant process observed in the crossed electron–molecular beam experiment. A and B denote reaction paths not observed in the experiment.

**Table 1 ijms-24-08706-t001:** The mass-to-charge ratio (*m*/*z*) of the fragment anions formed upon DEA to BrSU, including the peak positions, experimental onsets, and thermochemical thresholds obtained at the M062X/aug-cc-pVTZ level of theory. Superscripts in the column “Theory” correspond to reaction numbers in Figure 2.

Mass (*m*/*z*)	Anion	Peak Positions (eV)					Threshold (eV)	
		1	2	3	4	5	Exp.	Theory
206	BrSU^−^	≈0	0.55	--	--	--	≈0	−1.18 ^1^
205	(BrSU-H)^−^	≈0	0.2	0.5	0.8	1.3	≈0	−0.82 ^2a^ 1.54 ^2b^
127	(BrSU-Br)^−^	≈0	0.17	0.54	--	--	≈0	−1.03 ^3a^ 1.17 ^3b^ −0.44 ^3c^
126	(BrSU-HBr)^−^	≈0	0.25	0.7	--	--	≈0	0.21 ^4a^ 0.70 ^4b^ 1.27 ^4c^
79	Br^−^	≈0	0.4	1.0	2.3	4.8	≈0	0.13 ^5^
42	CNO^−^	≈0	1.0	2.5	3.4	4.6	≈0	1.13 ^6a^ 1.30 ^6b^
33	SH^−^	0.1	1.2	2.4	5.1	--	≈0	1.74 ^7^

**Table 2 ijms-24-08706-t002:** Thermodynamic thresholds calculated at the M062X/aug-cc-pVTZ-level for the formation of possible fragment anions resulting from DEA to BrSdU. Superscripts in the column “Theory” correspond to reaction numbers in Figure 5.

Mass (*m*/*z*)	Anion	T (K)	Threshold (eV) Theory
205	(BrSdU-deoxyribose)^−^	385.15	−0.34 ^1^
190	(BrSdU-deoxyribose-NH)^−^	385.15	1.40 ^2^
127	(BrSdU-deoxyribose-Br+H)^−^	387.15	0.45 ^3a^ 0.66 ^3b^ 1.68 ^3c^
126	(BrSdU-deoxyribose-Br)^−^	387.15	0.31 ^4a^ 1.23 ^4b^
98	(C_5_H_6_O_2_)^−^	385.15	2.71 ^5^
79	Br^−^	387.15	0.14 ^6^
42	NCO^−^	387.15	1.15 ^7a^ 2.48 ^7b^
33	SH^−^	387.15	1.70 ^8^

## Data Availability

The data presented in this study are available in the Appendix A.

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
