# Peer review of "Electron-Induced Decomposition of 5-Bromo-4-thiouracil and 5-Bromo-4-thio-2′-deoxyuridine: The Effect of the Deoxyribose Moiety on Dissociative Electron Attachment"

_ijms, 2023, doi:10.3390/ijms24108706_

Round 1

Reviewer 1 Report

In this work, Izadi et al. are studying by means of computations methods and experimental techniques the dissociative electron attachment phenomenon in two bromine derivatives of thiouracil, with and without deoxyribose. The study contains some interesting data and relevant conclusions in the field of the radiotherapy. Therefore, I recommend publication of the work once the following issues are addressed:

-        At the beginning of section 2, the authors write “According to our calculations, the electron affinity (EA) 114 of BrSU is positive and amounts to 1.18 eV”. There are other molecules with the same situation (less stable anion). As mentioned by the authors, in molecules in which the anion is less stable than the neutral, two types of anions can be formed, valence-bond and dipole-bond anions. The former is much higher in energy and for DEA, the latter seems to be more important according to the state-of-the-art based on accurate computational studies. To properly characterize and study dipole-bond anions, especial basis sets must be used. Otherwise, the solution is totally non-reliable. It is a spurious solution in which the electron is located in the more diffused orbital allowed by the basis set. With conventional basis sets, by changing the basis set, change totally the energy. Some discussion and citations in this context would enrich the text. It could be also helpful to verify if some of the anionic wavefunctions correspond to such “false” solutions by the inspection of the singly occupied molecular orbital (SOMO).  

-          In the reaction scheme (3), the nomenclature for BrSU-Br is not very clear. It looks like there 3 Br atoms in the fragment. The nomenclature could be explained at the beginning or somewhere.

-          On page 5, it is written that “This process is related to proton transfer from 168 N1-H to the C5 site in the fragment anion forming due to the bromine atom release 169 (Figure 2, reaction 3a), as suggested for BrU in Ref. [45].” How does this proton transfer take place? Any idea for the mechanism? This transfer of proton is expected to have a large energy barrier or barriers involved. Where is the energy coming from if it occurs at very low electron energies?

-          Why enthalpies are computed rather than Gibbs free energies?

-          Since some of the systems are open-shell radicals, are the authors using unrestricted-DFT?

Author Response

See word file

Author Response

See word file

Round 2

Reviewer 1 Report

In the revised manuscript, the authors have addressed the issues arisen by the Reviewer. The responses are in most of the cases properly clarifying the doubts. However, in the implementation of the changes in the manuscript there are some deficiencies. This must be improved.

-          To answer one of the question, the authors have added the text “In the reaction studied with the experiment electron energy is instantaneously absorbed by a substrate moving the system from the enthalpy level of the substrate to that of the products.” The meaning here seems to be that upon electron attachment, the systems goes directly to the products, without passing by an intermediate anion of the substrate previous to the dissociation. This must be clarified.  

-          In relation to the previous issue, it is also written that “This is why computational reaction enthalpies rather than reaction free enthalpies should be compared with the experimental thermodynamic thresholds.” This sentence is not clear to me.

-          Regarding the issue corresponding to the proton transfer mechanism occurring in reaction 3a, the authors are performing several calculations of reaction paths, introducing new data in the discussion text in the manuscript. However, the details of such calculations and related information are not added to the Methodological section. It should be added to properly explain how all the data in the text was obtained, and for reproducibility purposes.

These are my recommendations to clarify the minor last aspects of the work. Once it is done, I recommend publication of the manuscript.

Author Response

Please see file attached
